# STN-DBS Induces Acute Changes in β-Band Cortical Functional Connectivity in Patients with Parkinson’s Disease

**DOI:** 10.3390/brainsci12121606

**Published:** 2022-11-23

**Authors:** Matteo Conti, Alessandro Stefani, Roberta Bovenzi, Rocco Cerroni, Elena Garasto, Fabio Placidi, Claudio Liguori, Tommaso Schirinzi, Nicola B. Mercuri, Mariangela Pierantozzi

**Affiliations:** 1Parkinson Centre, Department of Systems Medicine, University of Rome “Tor Vergata”, 00133 Rome, Italy; 2Neurology Unit, Department of Systems Medicine, University of Rome “Tor Vergata”, 00133 Rome, Italy

**Keywords:** Parkinson’s disease, DBS, EEG-based functional connectivity

## Abstract

Subthalamic nucleus deep-brain stimulation (STN-DBS), in addition to a rapid improvement of Parkinson’s disease (PD) motor symptoms, can exert fast, local, neuromodulator activity, reducing β-synchronous oscillations between STN and the motor cortex with possible antikinetic features. However, STN-DBS modulation of β-band synchronization in extramotor cortical areas has been scarcely explored. For this aim, we investigated DBS-induced short-term effects on EEG-based cortical functional connectivity (FC) in β bands in six PD patients who underwent STN-DBS within the past year. A 10 min, 64-channel EEG recording was performed twice: in DBS-OFF and 60 min after DBS activation. Seven age-matched controls performed EEG recordings as the control group. A source-reconstruction method was used to identify brain-region activity. The FC was calculated using a weighted phase-lag index in β bands. Group comparisons were made using the Wilcoxon test. The PD patients showed a widespread cortical hyperconnectivity in β bands in both DBS-OFF and -ON states compared to the controls. Moreover, switching on STN-DBS determined an acute reduction in β FC, primarily involving corticocortical links of frontal, sensorimotor and limbic lobes. We hypothesize that an increase in β-band connectivity in PD is a widespread cortical phenomenon and that STN-DBS could quickly reduce it in the cortical regions primarily involved in basal ganglia–cortical circuits.

## 1. Introduction

Parkinson’s disease (PD) is a neurodegenerative disorder, primarily characterized by the degeneration of dopaminergic neurons of the substantia nigra. Nevertheless, it is now widely recognized that PD is a multisystem disease, involving several subcortical and cortical structures, possibly reflecting an ascending progression of synucleinopathy [1], even if a top-down cortical pathogenesis of PD has been recently proposed [2]. Moreover, neuroinflammation is known to play a fundamental role in the multidimensional pathogenesis of PD, as suggested by several recent studies [3,4,5]. In accordance, previous data, mainly based on functional magnetic resonance imaging (fMRI), have documented cortical functional connectivity (FC) deficits in patients with PD [6], suggesting that FC changes may concur to clinical manifestations of PD.

Bilateral subthalamic nucleus (STN) deep-brain stimulation (DBS) is a successful surgical treatment in patients with advanced PD [7]. Aside from the well-documented clinical and electrophysiological effects related to chronic stimulation, STN-DBS can exert a rapid improvement in PD motor symptoms, associated to a fast neuromodulator activity, with rapid local changes in the β band [8,9,10]. In particular, STN-DBS acutely reduces β-synchronous oscillations, which are collected by local field potentials (LFPs), linking STN and globus pallidus internal nucleus (GPi) as well as these basal ganglia (BG) structures and motor cortical areas, especially the supplementary motor area [11].

Previous studies have suggested that BG and motor-region synchronization in β bands is a typical feature of PD, playing a pivotal role in disease pathogenesis and akinetic symptoms [12]. However, the exact pathophysiology of this phenomenon is still unknown. It has been speculated that β-band synchronization between the BG and the cortex may be driven from the cortical motor areas [11], although the involvement of extramotor cortical areas has been scarcely explored. In this context, we investigated the possible short-term effects of STN-DBS on the entire corticocortical FC in β bands by means of a high-density EEG.

## 2. Materials and Methods

In the current study, we included patients diagnosed with idiopathic PD, according to the MDS clinical diagnostic criteria [13], who followed up at the neurological clinic of the University “Tor Vergata”, Rome, and who underwent STN-DBS bilateral implantation within the past 12 months. Patients participating in the study met the following entry criteria: (1) optimization of DBS parameters; (2) no comorbid major medical disorders and/or concomitant psychiatric disorders; (3) no cognitive impairment, as quantified by a Mini Mental State Examination score > 24.

We also enrolled a healthy control (HC) group of volunteers, who were similar in age and sex to the patients. The HC group was composed of subjects under EEG and brain MRI scrutiny as part of diagnostic tests, which excluded epilepsy or other neurological diseases. The entry criteria for the HC were: (1) no major medical disorders; (2) no neurological and/or psychiatric disorders; (3) no history of epilepsy or other conditions that could justify alterations in the EEG; (4) no cognitive impairment, as quantified by a Mini Mental State Examination score > 24.

To investigate the short-term effect of DBS on cortical FC of patients, a 10 min high-density EEG recording was performed twice: the first in STN-DBS-OFF, after turning off the DBS the night before (12 h before) and 12 h after the last administration of antiparkinsonian therapy; the second in STN-DBS-ON, one hour after turning on the DBS. Patient motor disability was scored using the Unified Parkinson’s Disease Rating Scale (UPDRS) part III at the same time as the EEG recordings. A single identical 10 min EEG recording was made in the HC.

EEG recording was executed in awake-resting state: the subjects were instructed to keep the eyes closed while staying awake. EEG data were recorded at a sampling rate of 1024 Hz, band-pass filtered at 0.5–50 Hz using a 64-channel EEG system (EbNeuro BePlus Pro Standard). Scalp electrodes were positioned according to 10-10 International System [14]. Impedance was kept below 30 kΩ.

A cleaning algorithm for EEG data was used [15]. We applied detrending and re-referencing to each channel, and a notch filter at 50 Hz. Independent component analysis (ICA) was used to remove EEG artifacts [16].

After EEG recording and removal of artifacts, we proceeded to the EEG source localization, which consists of two steps, solving forward and inverse problems. For the first step, we used a boundary element method (BEM) [17]: personal MRI and EEG data were coregistered through identification of the same anatomical landmarks (left and right preauricular points and nasion). Head model was computed by segmented MRI using CAT12 software [18]. In the second step, we used weighted minimum-norm estimation (wMNE) to solve the inverse problem [19]. The obtained sources were divided into 68 brain regions, using the Desikan–Killiany atlas [20].

The power spectral density was computed in β frequency bands (13–30 Hz), applying the Welch’s method. In the present study, the EEG data were divided into segments of 1 s in length, with an overlap of 50%.

FC was calculated using the values of weighted phase-lag index (wPLI), a measure derived from imaginary parts of coherence [21], between any pair of brain regions in β frequency bands, on segments of 1 s in length and with an overlap of 50%, according to Welch’s method. Analysis was made using Brainstorm toolbox for MATLAB R2022a [22].

To analyze the cortical network properties, we computed the mean connectivity matrices of wPLI of all subjects and calculated a cut-off value, so that 30% of all edges were considered significant. We used a threshold for individual connectivity matrices by absolute weight, in order to generate undirected networks. [23]. Degrees of single nodes were analyzed [23].

We analyzed the differences in the magnitude of power spectral density (i.e., sqrt (power)) in functional connectivity and network measure in β frequency bands between healthy controls and PD patients in STN-DBS-OFF and STN-DBS-ON. Group comparisons were made using the Wilcoxon rank test. Statistical analysis was performed using MATLAB R2022a (MathWorks, Inc., Natick, MA, USA).

## 3. Results

### 3.1. Subjects

We enrolled six patients with PD who met the entry criteria of the study. Seven healthy subjects constituted the HC group. The demographics and clinical characteristics of both groups are summarized in Table 1.

As expected, switching DBS on significantly improved motor symptoms (UPDRS part III DBS-OFF 56.0 ± 5.29 DBS-ON 24.0 ± 6.15, *p* < 0.001). Moreover, we analyzed specific subitems of UPDRS part III in the PD group in both DBS-OFF and DBS-ON states and found statistically significant differences. We found significant differences in rigidity (DBS-OFF 11.47 ± 2.38 DBS-ON 5.65 ± 1.13, *p* < 0.05), bradykinesia (DBS-OFF 21.67 ± 1.86 DBS-ON 9.33 ± 1.63, *p* < 0.001), tremors (DBS-OFF 9.5 ± 1.87 DBS-ON 2.17 ± 1.83, *p* < 0.001) and gait (DBS-OFF 6.67 ± 0.82 DBS-ON 2.5 ± 0.84, *p* < 0.001), but no difference was found in postural stability (DBS-OFF 3.33 ± 1.51 DBS-ON 2.5 ± 1.05, *p* = 0.29).

### 3.2. Spectral Analysis

The spectral analysis documented no significant differences between HC and PD patients, these latter considered either in STN-DBS-OFF or STN-DBS-ON. Within the PD patient group, a significant spectral power reduction was found in STN-DBS-ON compared to STN-DBS-OFF only in the following ROIs: R entorhinal (STN-DBS-OFF 34.42 ± 5.57 × 10^−23^ vs. STN-DBS-ON 0.67 ± 0.42 × 10^−23^, *p* < 0.05), R fusiform (STN-DBS-OFF 28.51 ± 18.11 × 10^−23^ vs. STN-DBS-ON 1.68 ± 1.20 × 10^−23^, *p* < 0.05), L inferotemporal (STN-DBS-OFF 37.98 ± 41.83 × 10^−23^ vs. DBS-ON 9.68 ± 1.46 × 10^−23^, *p* < 0.05), L postcentral (STN-DBS-OFF 42.33 ± 30.83 × 10^−23^ vs. DBS-ON 4.04 ± 2.98 × 10^−23^, *p* < 0.05) and L supramarginal (STN-DBS-OFF 41.73 ± 30.22 × 10^−23^ vs. STN-DBS-ON 0.48 ± 0.43 × 10^−23^, *p* < 0.05).

Comparisons in the spectral analysis of the β bands between PD patients and HCs are reported in Figure 1. The analysis was carried out at the level of the cortical sources. Spectral density power of current dipoles is expressed in A^2^m^2^/Hz.

### 3.3. Functional Connectivity Analysis

A comparison between HC and PD patients in STN-DBS-OFF showed a widespread, significant increase in β cortical FC in PD patients, involving intra- and interlinks of all cerebral lobes. We also found a significant increase in β FC in STN-DBS-ON compared to HC, which, contrary to what was observed in STN-DBS-OFF, did not include the following functional blocks: intrafrontal mean connectivity, frontotemporal, frontolimbic and frontoparietal mean connectivity (Figure 2).

Within the PD group, we observed a reduction in β FC after STN-DBS was turned on compared to STN-DBS-OFF; specifically, block-based statistical analysis revealed a significant difference in mean connectivity (measured in wPLI) in intrafrontal, intrasensorimotor, intratemporal, intralimbic, frontosensorimotor, frontotemporal, frontolimbic, temporolimbic and temporoparietal areas (Figure 2, Table 2).

### 3.4. Network Measures

Compared to HC, we documented a significant increase in mean lobar nodal degrees based on β FC in PD patients in both STN-DBS-OFF (frontal, sensorimotor, temporal, limbic, parietal and occipital lobes) and STN-DBS-ON (sensorimotor, temporal, parietal and occipital lobes). Within the PD group, under the condition STN-DBS-ON, PD patients showed a significant reduction in mean lobar nodal degrees based on β FC compared to STN-DBS-OFF, but only in frontal and limbic nodes. A reduction not statistically significant in nodal degrees was also observed in sensorimotor, temporal, limbic and parietal areas. The nodal properties of the groups are summarized in Figure 3 and Table 3.

## 4. Discussion

In our study we documented that PD patients implanted with bilateral STN-DBS were characterized by an increase in cortical FC in β bands in DBS-OFF compared to controls in all cortical functional blocks. Moreover, the same patients, in DBS-ON, still showed β-band hyperconnectivity compared to controls in all blocks expected for intrafrontal, frontotemporal, frontolimbic and frontoparietal blocks. Consistently, switching on STN-DBS determined the acute reduction in cortical FC in β frequencies, primarily involving intra- and intercorticocortical connections of frontal, sensorimotor and limbic lobes.

Previous electrophysiological studies, focused on intraoperative LFP recordings from depth electrodes in patients with PD undergoing DBS, allowed the identification of an abnormally high β-power spectral density in the STN [12]. Furthermore, microelectrode studies in PD patients have also demonstrated synchronization of single units in β bands in the STN [24], while macroelectrode studies have shown β-synchronous oscillations between STN, GPi and motor cortical areas [25]. It has been found that in PD coherence between STN, GPi and motor cortex appears to decrease during movement preparation and execution [26], and under levodopa therapy [27]; hence, β-band synchronization may have antikinetic properties. Although the pathophysiological mechanism underlying the phenomenon of β-band synchronization is not fully understood, some studies support the hypothesis that β synchronization between cortical areas, STN and GPi is likely to be driven from the motor cortex [11,26].

On the other hand, synchronous oscillations in γ frequency bands between STN, GPi and motor areas have been observed in PD patients after levodopa administration [28]. The higher coherence in the γ frequency band between the cortex, STN and GPi is found at around 60–80 Hz, and at the double of this frequency. Contrary to what was observed for β synchronization, in this case, STN and GPi seem to drive γ cortical activity, with prokinetic effects [12]. Indeed, cortical γ oscillations have been positively related to the planning and executions of movements [29,30,31], but also with other brain functions, including sensory and cognitive processing, long-term memory and language tasks [32,33]. Consistently, an excessive increase in γ motor cortical activity has been correlated with the onset of levodopa-induced dyskinesias [34].

Therefore, β synchronization between BG and cortical areas in PD may rise accordingly to the reduction in prokinetic STN and GPi γ oscillations directed to cortical motor areas. Thus, dysfunction of the BG cortical circuit would trap cortical activity in a β-synchronous antikinetic pattern. Consistently, switching on STN-DBS reverts this phenomenon as patient bradykinesia is ameliorated.

Our results, showing an increase in β-band cortical FC in PD patients in DBS-OFF compared to controls in all cortical functional blocks, are in line with previous findings and, indeed, expand them; the hypothesis is that the increase in corticocortical β-band connectivity in PD is a widespread phenomenon, not limited to motor areas. Β-band cortical FC has been rarely studied in cortical areas other than sensorimotor, although it has been found that the increase in whole-brain β-band connectivity negatively influences audiovisual integration in old adults [35].

On the other hand, the selective reduction we observed in switching on DBS, characterized by a rapid decrease in β FC in frontal, sensorimotor and limbic areas, could be explained considering that STN is a structure centrally involved in motor, prefrontal and limbic BG cortical circuits. Conversely, persistence of β hyperconnectivity in the remaining cortical areas could be related to the impairment of other subcortical structures not modulated by STN-DBS. Degeneration of dopamine neurons in PD includes both the substantia nigra pars compacta and the ventral tegmental area, which project independently to the cortex via mesolimbic and mesocortical circuits. Moreover, it is well known that PD is a multisystem disease, with impairments in other ascendent monoaminergic systems [36]. However, we cannot exclude that β cortical hyperconnectivity might be due to a primary involvement of cortical neurons, considering the advanced state of the PD population of our study in relation to bottom-up model of disease progression [1]. Following this hypothesis, the selective modulatory effect of STN-DBS seems limited only to the BG cortical circuits, involving frontal, sensorimotor and limbic areas.

From a clinical point of view, it is of note that the analysis of specific UPDRS III subitems allowed us to identify significant differences not only in rigidity, bradykinesia and tremor domains, but also in the gait subitem. Therefore, we could hypothesize that the reduction in β FC in the frontal lobe could be related to gait improvement, as reduction in sensorimotor areas could be linked to segmental motor improvement. Nevertheless, the size of study population does not allow regression analyses between connectivity and clinical data, which are required to demonstrate this hypothesis.

We are aware that our study presents some limitations due to the low number of subjects included. Yet, we preferred to limit the observation to a patient cohort sharing similar inclusion criteria, the same surgical room and analogous recalage (avoiding uncertainties implicit to multicenter studies). That said, these results need to be confirmed with a larger PD population. Moreover, we were limited in investigating high-γ-band FC, which was not easily achievable with scalp EEG recordings due to muscle artifacts. Indeed, the analysis of high-γ FC would allow a better understanding of band-specific connectivity phenomena in different conditions, such as in PD patients, changing from OFF to ON states after STN-DBS activation in relation to dyskinesias, which has been linked to abnormal γ cortical activity.

## 5. Conclusions

In our study, we documented that PD patients implanted with STN-DBS were characterized by an increase in β-band cortical FC in DBS-OFF compared to controls in all cortical functional blocks, which persisted in DBS-ON expected for intrafrontal, frontotemporal, frontolimbic and frontoparietal blocks. Consistently, switching on STN-DBS determined the acute reduction in cortical FC in β frequencies, primarily involving intra- and intercorticocortical connections of frontal, sensorimotor and limbic lobes.

We hypothesize that an increase in β-band connectivity is a widespread cortical phenomenon in PD and that STN-DBS could quickly reduce it in the corticocortical connections of frontal, sensorimotor and limbic lobes primary involved in BG cortical circuits.

Further studies are required to confirm this observation and investigate the potential role of β cortical connectivity reduction in adaptive DBS.

## Figures and Tables

**Figure 1 brainsci-12-01606-f001:**
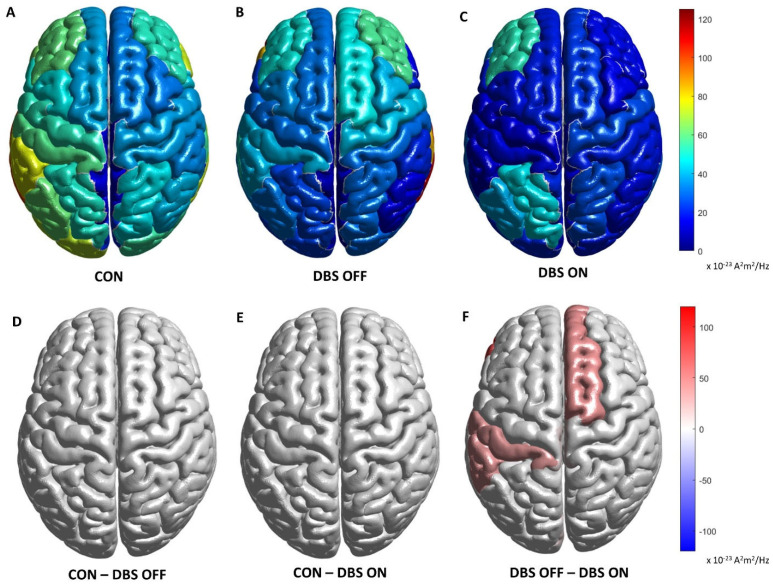
First row shows magnitude of power spectral density for each ROI, respectively, in controls (**A**) and in PD patients with DBS-OFF (**B**) and DBS-ON (**C**) states in β frequency bands. Second row reports differences in magnitudes of power spectral density between controls and PD DBS-OFF (**D**), controls and PD DBS-ON (**E**) and DBS-OFF and DBS-ON states (**F**) in β frequency bands. Only differences with a *p* < 0.05 are shown. CON: control group. DBS OFF: PD patients in DBS-OFF state. DBS ON: PD patients in DBS-ON state. CON–DBS OFF: difference between controls and PD patient in DBS-OFF state. CON–DBS ON: difference between controls and PD patient in DBS-ON state. DBS OFF–DBS ON: difference between PD patient in DBS-OFF and DBS-ON states.

**Figure 2 brainsci-12-01606-f002:**
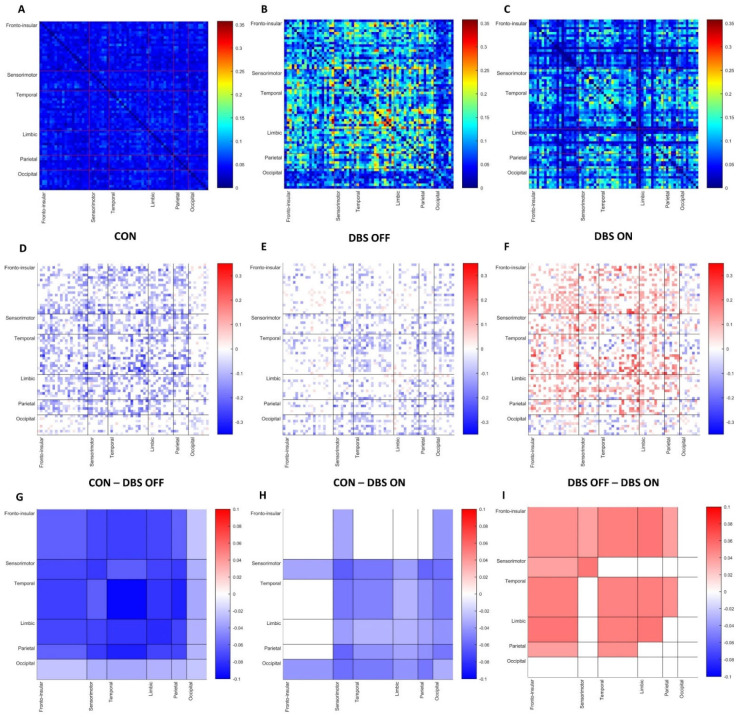
First row shows FC for each pair of ROIs based on wPLI, respectively, in HC (**A**) and PD patients STN-DBS-OFF (**B**) and STN-DBS-ON (**C**) in β frequency bands. Second row reports differences in FC of each pair of ROIs between HC and PD patients STN-DBS-OFF (**D**), HC and patients STN-DBS-ON (**E**) and between patients STN-DBS-OFF and STN-DBS-ON (**F**) in β frequency bands. Third row shows block-based differences in FC between HC and PD STN-DBS-OFF (**G**), HC and PD STN-DBS-ON (**H**) and between PD patients STN-DBS-OFF and STN-DBS-ON (**I**) in β frequency bands. Only differences with a *p* < 0.05 are shown.

**Figure 3 brainsci-12-01606-f003:**
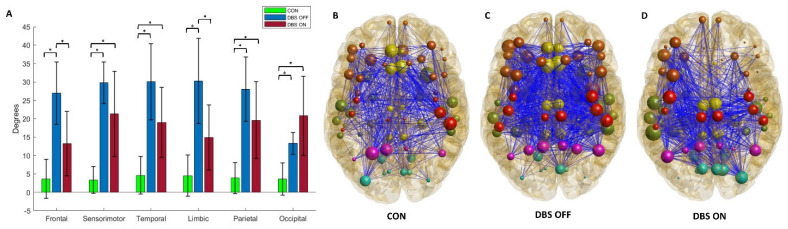
Box graph (**A**) of nodal degrees of each brain lobe in HC and PD patients in STN-DBS-OFF and STN-DBS-ON states. * indicates a statistically significant difference (*p* < 0.05). Graph representation of whole-brain network based on wPLI in β frequencies in HC (**B**) and PD patients in STN-DBS-OFF (**C**) and STN-DBS-ON (**D**).

**Table 1 brainsci-12-01606-t001:** Demographic and clinical characteristics of PD patients and healthy controls. Data are means ± standard deviation. UPDRS III: Unified Parkinson’s Disease Rating Scale part III (motor symptoms).

	PD	HC
N	6	7
Sex (M/F)	4/2	4/3
Age (years)	55.5 ± 4.23	52.3 ± 6.75
Disease Duration (months)	120 ± 20.82	
UPDRS III in STN-DBS-OFF	56.0 ± 5.3	
UPDRS III in STN-DBS-ON *	24.0 ± 6.2	

* UPDRS III examined 60 min after DBS was switched on.

**Table 2 brainsci-12-01606-t002:** Significant differences in block-based mean β FC between DBS-OFF and DBS-ON conditions.

Block	DBS-OFF	DBS-ON	*p*-Value
Intrafrontal	0.103 ± 0.017	0.059 ± 0.020	<0.005
Intrasensorimotor	0.124 ± 0.004	0.109 ± 0.038	<0.05
Intratemporal	0.144 ± 0.050	0.095 ± 0.018	<0.05
Intralimbic	0.129 ± 0.049	0.075 ± 0.023	<0.05
Frontosensorimotor	0.124 ± 0.004	0.109 ± 0.038	<0.05
Frontotemporal	0.119 ± 0.036	0.069 ± 0.020	<0.05
Frontolimbic	0.116 ± 0.030	0.062 ± 0.018	<0.005
Temporolimbic	0.128 ± 0.046	0.078 ± 0.020	<0.05
Temporoparietal	0.132 ± 0.035	0.088 ± 0.024	<0.05

FC is based on wPLI measure. Data are means ± standard deviation.

**Table 3 brainsci-12-01606-t003:** Lobar nodal degrees in controls and PD patients, in both DBS-OFF and DBS-ON.

Lobe	CON	DBS-OFF	DBS-ON	*p*-Value *	*p*-Value#	*p*-Value°
Frontal	3.64 ± 5.33	26.93 ± 8.45	13.23 ± 8.74	<0.005	NS	<0.05
Sensorimotor	3.35 ± 3.65	29.77 ± 5.66	21.35 ± 11.58	<0.005	<0.05	NS
Temporal	4.61 ± 5.13	30.02 ± 10.39	19.00 ± 9.54	<0.005	<0.05	NS
Limbic	4.52 ± 5.59	28.03 ± 8.79	14.90 ± 8.88	<0.005	NS	<0.05
Parietal	3.86 ± 4.25	19.61 ± 10.48	19.61 ± 10.48	<0.005	<0.05	NS
Occipital	3.60 ± 4.38	13.31 ± 2.95	20.79 ± 10.76	<0.005	<0.05	NS

*p*-Values * indicates a comparison between control group and PD DBS-OFF; *p*-Value# indicates a comparison between control group and PD DBS-ON; *p*-Value° indicates a comparison between PD DBS-OFF and DBS-ON. NS: Not significant difference.

## Data Availability

The datasets generated during the analysis are available from the corresponding author on reasonable request.

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
