# Peer review of "STN-DBS Induces Acute Changes in β-Band Cortical Functional Connectivity in Patients with Parkinson’s Disease"

_brainsci, 2022, doi:10.3390/brainsci12121606_

Round 1
Reviewer 1 Report
Over all it's a interesting study, with a detail explination about methods and the posterior data analysis. Anyhow, it would be more informative, if the autors could analize if there is a relation with the decrease in beta band and specific subitems of UPRDSIII like walk or freezing, in the change from SNT-DBS OFF to SNT-DBS ON state, for some cortico-cortical circuits like frontal afferents.
Would be interesting also compare this reduction in the beta band (from OFF to ON) in specific areas in relation to cognitive test, even in the abcense of cognitive impairment, scales like PD-CRS could diferenciate fronto-subcortical from posterior cortex. In this same way diferrences in this aspect between SNT_DBS OFF and HC would be also interesting, making remarkable the clincial implication of this findings in beta activity in cortical regions, for better understanding of the underlyeing mecanism. This could the inforce hypotesis of withspread brain degeneration in PD patients, and the implication in other symtomps futher the motor ones.
The autors also comment the limitation to quantificate gamma band, this could also some interest in the way to compare this changes (in beta and gamma band) in patients changing from OFF to ON state after applicated STN-DBS in relation with dyskinesias.
Author Response
We really appreciate your suggestions that improve the manuscript. We tried to answer all your questions and we changed part of the text.
As suggested, we added analysis of specific subitems of UPDRS III in results. We also discuss these findings in the discussion.
As regards the second point, unfortunately in the present study specific tests for cognitive impairment, such as PD-CRS, were not performed. Therefore, we cannot discuss possible cognitive implications of the abnormal beta connectivity we observed in frontal regions. However, we really thank you for the suggestion that will surely help to improve a future more extensive work.
Finally, as suggested, we added some considerations about gamma cortical activity in relation to dyskinesias into the discussion and the possibility of better understand this phenomenon by means of connectivity analyses.
Reviewer 2 Report
beta-synchronisation and its putative consequence for structutre and function in PD perhaps may be discussed further
gamma-oscillation - much the same.
Author Response
We really appreciate your suggestions. We tried to expand the discussion on clinical and pathological consequence of gamma and beta connectivity modifications in PD patients.
Reviewer 3 Report
Authors of the work by Stefani et al present an overview on beta-cortical functional connectivity. The main outcomes of the study were:
1. PD patients showed a widespread cortical hyperconnectivity in β band in both DBS-OFF and ON states 19 compared to controls.
2. Switching-on STN-DBS determined an acute reduction of β FC, 20 primarily involving corticocortical links of frontal, sensorimotor and limbic lobes.
The work is of interest nad presents a relevant aspects of PD. I have several comments, which could additionally improve the work:
1. In the introduction the context of the pathogenesis of PD is too vague. Though authors state that this is a multisystem neurodegenerative disease, it would be valuable to indicate the hypothesised mechanisms of the processes as the multidimensional inflammatory processes
Ref.
Microglia and astrocyte activation is region-dependent in the α-synuclein mouse model of Parkinson's disease. Glia. 2022 Nov 10. doi: 10.1002/glia.24295. Epub ahead of print. PMID: 36353934.
Platelet-to-lymphocyte ratio and neutrophil-tolymphocyte ratio may reflect differences in PD and MSA-P neuroinflammation patterns. Neurol Neurochir Pol. 2022;56(2):148-155. doi: 10.5603/PJNNS.a2022.0014. Epub 2022 Feb 4. PMID: 35118638.
2. In the material section mention the exclusion criteria:
"no comorbid medical disorders and/or concomitant 55 psychiatric disorders"
This issue should be presented in a more precise way e.g. were patients with hypertension or minor comorbidites not impacting the results of the study also excluded?
3. Adding a table with an overview of the obtained results would be valuable.
Author Response
We really appreciate your suggestions that improve the manuscript. We tried to answer all your questions, we changed part of the text and added two tables, summarizing results of the study.
Responses to your comments follow:
- As suggested, we have extended the introductory part on the pathogenesis of PD, including neuroinflammation in the multidimensional pathological process of the disease;
- we added the adjective major to medical disorder, as hypertension or minor comorbidities were not exclusion criteria for the study;
- As suggested, we added two tables that summarized the obtained results.